# Cerebral Expression of Metabotropic Glutamate Receptor Subtype 5 in Idiopathic Autism Spectrum Disorder and Fragile X Syndrome: A Pilot Study

**DOI:** 10.3390/ijms22062863

**Published:** 2021-03-11

**Authors:** James Robert Brašić, Ayon Nandi, David S. Russell, Danna Jennings, Olivier Barret, Samuel D. Martin, Keith Slifer, Thomas Sedlak, John P. Seibyl, Dean F. Wong, Dejan B. Budimirovic

**Affiliations:** 1Section of High Resolution Brain Positron Emission Tomography Imaging, Division of Nuclear Medicine and Molecular Imaging, The Russell H. Morgan Department of Radiology and Radiological Science, The Johns Hopkins University School of Medicine, Baltimore, MD 21287, USA; anandi1@jh.edu (A.N.); smart149@jhu.edu (S.D.M.); tsedlak@jhmi.edu (T.S.); dfwong@wustl.edu (D.F.W.); 2Institute for Neurodegenerative Disorders, New Haven, CT 06510, USA; drussell@invicro.com (D.S.R.); Jennings@dnli.com (D.J.); olivier.barret@cea.fr (O.B.); jseibyl@invicro.com (J.P.S.); 3Research Clinic, Invicro, New Haven, CT 06510, USA; 4Denali Therapeutics, Inc., South San Francisco, CA 94080, USA; 5Laboratoire des Maladies Neurodégénératives, Molecular Imaging Research Center (MIRCen), Institut de Biologie François Jacob, Centre National de la Recherche Scientifique (CNRS), Commissariat à l’Énergie Atomique et aux Énergies Alternatives (CEA), Université Paris-Saclay, 92265 Fontenay-aux-Roses CEDEX, France; 6Department of Neuroscience, Zanvyl Krieger School of Arts and Sciences, The Johns Hopkins University, Baltimore, MD 21218, USA; 7Department of Psychiatry and Behavioral Sciences-Child Psychiatry, The Johns Hopkins University School of Medicine, Baltimore, MD 21205, USA; slifer@kennedykrieger.org; 8Department of Behavioral Psychology, Kennedy Krieger Institute, Baltimore, MD 21205, USA; 9Department of Psychiatry and Behavioral Sciences-General Psychiatry, The Johns Hopkins University School of Medicine, Baltimore, MD 21205, USA; 10Laboratory of Central Nervous System (CNS) Neuropsychopharmacology and Multimodal Imaging (CNAMI), Mallinckrodt Institute of Radiology, Washington University, Saint Louis, MO 63110, USA; 11Department of Psychiatry, Kennedy Krieger Institute, Baltimore, MD 21205, USA

**Keywords:** binding potential, cortex, caudate nucleus, cingulate, fragile X mental retardation 1 gene (*FMR1*), neurodevelopmental disorders, positron emission tomography (PET), putamen, radiotracer, thalamus

## Abstract

Multiple lines of evidence suggest that dysfunction of the metabotropic glutamate receptor subtype 5 (mGluR_5_) plays a role in the pathogenesis of autism spectrum disorder (ASD). Yet animal and human investigations of mGluR_5_ expression provide conflicting findings about the nature of dysregulation of cerebral mGluR_5_ pathways in subtypes of ASD. The demonstration of reduced mGluR_5_ expression throughout the living brains of men with fragile X syndrome (FXS), the most common known single-gene cause of ASD, provides a clue to examine mGluR_5_ expression in ASD. We aimed to (A) compare and contrast mGluR_5_ expression in idiopathic autism spectrum disorder (IASD), FXS, and typical development (TD) and (B) show the value of positron emission tomography (PET) for the application of precision medicine for the diagnosis and treatment of individuals with IASD, FXS, and related conditions. Two teams of investigators independently administered 3-[^18^F]fluoro-5-(2-pyridinylethynyl)benzonitrile ([^18^F]FPEB), a novel, specific mGluR_5_ PET ligand to quantitatively measure the density and the distribution of mGluR_5_s in the brain regions, to participants of both sexes with IASD and TD and men with FXS. In contrast to participants with TD, mGluR_5_ expression was significantly increased in the cortical regions of participants with IASD and significantly reduced in all regions of men with FXS. These results suggest the feasibility of this protocol as a valuable tool to measure mGluR_5_ expression in clinical trials of individuals with IASD and FXS and related conditions.

## 1. Introduction

Autism spectrum disorder (ASD) [1] comprises a heterogeneous group of neurodevelopmental disorders including (A) a subtype characterized by a behavioral phenotype with no known etiology [idiopathic autism spectrum disorder (IASD)] [2] and (B) medical disorders with known genetic etiologies [3], such as fragile X syndrome (FXS) [4]. All subtypes of ASD are characterized by impaired social communication and repetitive and restricted behaviors and interests [1,2,3,4,5]. Additionally, FXS and some other subtypes of ASD are also characterized by intellectual disability (ID) [6]. Dysfunction of protein synthesis mediated by abnormal pathways including metabotropic glutamate receptors (mGluR) plays a role in the pathometabolism of IASD [7,8] and FXS [9,10,11,12]. Despite the evidence for dysfunction of mGluR_5_ expression in IASD, conflicting findings include the decreased expression in the dorsolateral prefrontal cortex [13] and increased expression in the post-central gyrus and the cerebellum [14].

The confusion about mGluR_5_ expression in IASD may be resolved utilizing techniques that have provided convergent validity to studies of mGluR_5_ expression in FXS, the most common single-gene cause of ASD and ID. FXS results from the presence of the fragile X mental retardation 1 (*FMR1*) gene leading to deficits of Fragile X Mental Retardation Protein (FMRP). Dysregulated activation of group I metabotropic glutamate receptors [metabotropic glutamate receptors subtypes 1 and 5 (mGluR_1/5_)] causing metabotropic glutamate receptor dependent long-term depression (mGluR-LTD) plays a role in the pathogenesis of FXS [15,16]. The mechanisms of mGluR_1/5_ dysregulation leading to the neurobehavioral symptoms of FXS have been elucidated by the study of *fmr1* knockout (KO) mouse models. The deficits of FMRP in *fmr1* KO mouse models result in dysfunction of crucial group 1 metabotropic glutamatergic pathways leading to dysregulated downstream signaling cascades including the mammalian target of rapamycin (mTOR) and the mitogen-activated protein kinase (MAPK) extracellular signal-regulated kinase (ERK) pathways [17]. The correction of mGluR-LTD and behavioral symptoms in *fmr1* KO mouse models suggests that the a biomarker to measure mGluR_5_ expression in the living human brain represents a means to apply precision molecular medicine to ameliorate behavioral symptoms of FXS and possibly other subtypes of ASD [9,10,17,18,19,20,21,22,23,24].

Clinical trials of FXS have been flawed by several limitations, including the absence of a tool to measure the expression of mGluR_5_ in the living brains of participants with FXS [9,12,17,24]. We showed that 3-[^18^F]fluoro-5-(2-pyridinylethynyl)benzonitrile ([^18^F]FPEB), a novel, specific mGluR_5_ ligand to quantitatively measure the density and distribution of mGluR_5_s in the brain regions of humans through PET (Figure 1) [25] may be a promising means to obtain quantitative measurements of mGluR_5_ expression in individuals with IASD or FXS for clinical trials and other investigations [14,17,23,26]. We seek to expand our investigations to compare and contrast mGluR_5_ expression for participants of both sexes with IASD [14,23] and typical development (TD) and men with FXS [17,24,26].

Development of interventions to ameliorate the specific molecular deficits of individuals with IASD and FXS with and without ASD will facilitate the utilization of precision medicine to target the unique needs of each person [9,24].

We aimed to (A) compare and contrast mGluR_5_ expression for participants of both sexes with IASD [14,23] and typical development (TD) [14,25] and men with FXS [17,26] and (B) show the value of PET with ([^18^F]FPEB) for the application of precision medicine for the diagnosis and treatment of individuals with IASD, FXS, and related conditions [9,17,24,28].

## 2. Results

The clinical characteristics of all participants {group (IASD, FXS, or TD), institution [Institute for Neurodegenerative Disorders (IND) or Johns Hopkins University (JHU)], sex (female or male), age in years, and basal metabolic index (BMI)} are tabulated in Table 1 [26,29].

The mGluR_5_ uptake of participants in the regions of interest (ROI) [caudate nucleus (CN), medial temporal cortex (mTp), occipital cortex (Oc), parietal cortex (Pa), posterior cingulate cortex (pCg), putamen (Pu), thalamus (Th), and temporal lobe (Tp)] of all participants are recorded in Table 2 [26,29].

The ages of participants with IASD were lower than those with FXS and TD (Table 1) [17,26,29]. BMIs were ordered IASD < TD < FXS (Table 1) [17,26,29].

mGluR_5_ uptake was ordered FXS < TD < IASD in cortical (Oc, Pa, Tp, and pCg) structures (Figure 2) [26,30].

By contrast mGluR_5_ uptake was lower in participants with FXS than in participants with TD and IASD in subcortical (CN, Pu, and Th) structures (Figure 3) [26,30].

The initial visual analysis of the data indicated that the mGluR_5_ uptake differed across the groups of FXS, IASD, and TD in multiple regions. Analysis of variance (ANOVA) confirmed that group had a significant effect across all regions (d.f. = 2, F = 51.6, *p* < 0.001) (Table 3) [30]. *Post hoc* pair-wise comparisons using Tukey’s Honest Standard Differences (HSD) method further confirmed specific differences (Table 4) [30,31]. The pairwise comparisons highlighted the largest group differences in the temporal cortex (adjusted mean difference, FXS versus IASD = −2.19 ± 0.49 (*p* < 0.001) the parietal cortex (FXS versus IASD = −2.31 ± 0.48, *p* < 0.001), and the occipital cortex (FXS versus IASD = −1.88 ± 0.41, *p* < 0.001) [30,31].

## 3. Discussion

We confirmed our earlier finding that mGluR_5_ expression is reduced in all brain regions in men with FXS [17] on a sample of men with FXS compared to participants of both sexes with IASD [14,23] and TD [14,17,25,26]. In men with FXS, reduced mGluR_5_ expression in (A) cortical regions provides a basis for ID and (B) limbic regions provides a basis for the neurobehavioral symptoms [10,17].

We expanded our finding of increased mGluR_5_ expression in the postcenteral gyrus and cerebellum of men with IASD [14] to show increased mGluR_5_ expression in cortical regions of a sample from two separate institutions (IND and JHU) that includes participants with IASD and TD of both sexes. There are several possible explanations for the opposite results in IASD versus FXS. First, there may be different characteristic mechanisms for the development of mGluR_5_ expression in IASD and FXS. Second, there may be other characteristics of these cohorts, specifically age and ID, that caused the differences in mGluR_5_ expression in the cohorts with FXS and IASD. The participants with IASD were all younger than the participants with FXS. There may be reductions in mGluR_5_ expression correlated with age as for dopamine D2 and serotonin S2 receptors [32]. Additionally all participants with IASD were recruited from samples with high-functioning autism; all participants with IASD had normal or superior intelligence. By contrast all participants with FXS had ID. Therefore, the opposite results of mGluR_5_ expression in IASD versus FXS may simply reflect the differences in age and ID between the cohorts. The opposite results of mGluR_5_ expression in IASD versus FXS may therefore be unrelated to the diagnosis of FXS and IASD.

These findings confirm the hypothesis that mGluR_5_ expression plays a role in the pathogenesis of FXS and other subtypes of IASD. The protocol for this investigation provides a feasibility tool that may facilitate the measurement of a biomarker of mGluR_5_ expression to conduct rigorously designed clinical trials of FXS [9] and perhaps other subtypes of IASD. That said, the findings of this study merit replication in a larger sample of the groups studied here and other neurodevelopmental disorders [33]. Indeed the current protocol may be expanded to promote knowledge about multiple neuromodulators in FXS, Rett syndrome [34,35] and other subtypes of IASD.

Limitations. Estimation of binding potential for participants from IND as [standard uptake value ratio (SUVR)-1] [36] introduced uncertainty in the analysis [17]. Additionally the comparison of results from IND and JHU was confounded by the use of differences in scanners, scanning times, and analysis [17,26]. The similarity of results from both IND and JHU suggests the presence of convergent validity that the findings represent the characteristics of the status (IASD, FXS, and TD) of the participants. Future investigations will be enhanced by contemporaneous conduct of all investigations at all participating institutions with identical protocols and analyses [17].

Additionally, since some participants with TD are much older than other participants, the age difference may represent a confounding influence. mGluR_5_ density may be reduced with age just as the density of dopamine D2 and serotonin S2 receptors is reduced with age [32]. The variability of BMIs may represent another confounding influence. Since all participants with IASD were high-functioning [14,23,26], both samples of participants with IASD and TD lacked the intellectual disability (ID) that characterized the sample of males with FXS [17,26].

Future directions. Our finding of increased mGluR_5_ in the post central gyrus and cerebellum of men with IASD [14] was expanded in the current report with a sample of participants with IASD and TD of both sexes and men with FXS. We confirmed the reductions in mGluR_5_ in all regions in men with FXS [17]. We showed increased mGluR_5_ in cortical regions of participants with IASD. A study of the left striatum of a different cohort of participants with ID and TD demonstrated a trend of increased mGluR_5_ in participants with IASD by PET with [^18^F]FPEB, no change in glutamate by magnetic resonance spectroscopy (MRS), a trend of decreased gamma amino butyric acid (GABA) by MRS, and a strong negative correlation between mGluR_5_ and GABA [37]. This finding supports the hypothesis of abnormal excitatory/inhibitory ratio in participants with IASD [2] and merits expansion and confirmation in other cohorts using both PET and MRS to assess both mGluR_5_ and GABA. 

Future investigations utilizing the protocol of this study may provide the tools for successful clinical trials of negative allosteric modulators (NAMs) for FXS and IASD. Despite the evidence that NAMs ameliorate behavioral symptoms in animal models of FXS, there have not been beneficial effects demonstrated in multiple clinical trials of NAMs in FXS. Flaws in the design of the clinical trials including the absence of a tool to measure mGluR_5_ expression in the living human brains of participants with FXS have been identified as likely explanations for the unsuccessful clinical trial of NAMs in FXS [9]. Therefore, utilization of the procedure in this study may provide the crucial tool to generate rigorous measurements to demonstrate beneficial neurobehavioral effects of NAMs in clinical trials of FXS and IASD and related conditions.

Additional investigations will be enhanced with multiple imaging techniques including PET, MRS, PET/MRI [38], electroencephalography (EEG) [39,40], event-related brain potential (ERP) [39,40,41], resting state functional magnetic resonance imaging (rs-fMRI), diffusion tensor imaging (DTI), movement measurements [42], and quantitative measurements of FMRP and the *FMR1* gene [43]. Further prospective studies of ASD [44] may be enhanced by including these measurement tools, neuropsychological assessments, and whole exome sequencing (WES) [45]. The evidence for decreased expression of FMRP in IASD [46] and FXS [43] indicates that correlation of FMRP with mGluR_5_ [43,46] and GABA in ASD [37] is appropriate for future studies.

## 4. Materials and Methods

### 4.1. Participants

#### 4.1.1. Recruiting Sites

The study is approved by Johns Hopkins Medicine Institutional Review Board IRB 169,249 [17]. The protocols for the study of humans were approved by the Institutional Review Boards of the Institute for Neurodegenerative Disorders (IND) in New Haven, Connecticut [47] and the Johns Hopkins University (JHU) in Baltimore, Maryland [48,49]. Since exposure to radioactivity in PET constitutes greater than minimal risk, this pilot study was restricted to adults [17]. Written informed consent was obtained from each participant at both locations.

We report the findings of cohorts of independent investigations conducted at the IND on seven men with FXS (mean age 27 ± 4.76, range 12 years) [17,26], one man with fragile X syndrome allele size mosaicism (FXS-M) aged 56.6 years [17,26], and five men and six women with TD (mean age 38.27 ± 15.68, range 42 years) [17,26], and at the JHU on two men with FXS (mean age 25.5 ± 2.12, range 3 years), six men and one woman with IASD (mean age 19.71 ± 2.06, range 5 years), and five men and two women with TD (mean age 26.57 ± 7.14 range 20 years) [14,17,23,25,26]. In contrast to the participants with FXS and FXS-M, all participants with IASD and TD had no evidence of intellectual disability (ID) [26]. In order to maximize the size effect, this report with focus on the combined sample of participants with IASD (*N* = 7, age 19.71 ± 2.06), FXS (*N* = 10, age 29.7 ± 10.39), and TD (*N* = 19, age 34.89 ± 14.57) [26,29].

#### 4.1.2. Inclusion Criteria

Inclusion criteria for all participants included age between 18–60 years. Participants with IASD had a diagnosis of autism based on the Autism Diagnostic Interview-Revised [14,23,50], the Autism Diagnostic Observation Schedule [14,23,51], the Diagnostic and Statistical Manual of Mental Disorders, Fifth Edition (*DSM-5*) [1,14], and other diagnostic tools documented in our prior publication [14,17,26]. Participants with FXS had a diagnosis of FXS based on *FMR1* DNA gene testing by polymerase chain reaction (PCR)/Southern Blot, supplemented by clinical neurobehavioral profiling [17,26,43].

#### 4.1.3. Exclusion Criteria

Exclusion criteria were clinically significant abnormal laboratory values and/or clinically significant unstable serious medical, neurological, or psychiatric illnesses [14,17,47].

### 4.2. Procedures

#### 4.2.1. Positron Emission Tomography (PET)

All participants underwent scans conducted by an experienced research staff of Certified Nuclear Medicine Technologists (CNMT) who had attained certification by the Nuclear Medicine Technology Certification Board (NMTCB). The technologists had conducted many PET scans before this study. The technologists maintained the physical conditions of each scan optimally for the completion of the scans. Participants were positioned by the technologists in the most comfortable manner for scans. Heads were stabilized in the scanner by gauge strips at IND and by face masks at JHU [17,34]. In order to maintain a comfortable environment during the scans, technologists utilized blankets and pads to raise legs. The physical conditions of the scans were maintained in optimal manners for participants by outstanding technologists.

Positron emission tomography (PET) after the intravenous bolus injection 185 MBq (5 mCi) of [^18^F]FPEB [14,17,23,26] was conducted at IND on an ECAT EXACT HR+ PET manufactured by Siemens/CTI (Knoxville, TN) [52] for 90–120 min after injection and at JHU on an ECAT high resolution research tomograph (HRRT) manufactured by Siemens/CTI (Knoxville, TN) [53] for 0–90 min after injection. Injectors obtained measured doses of [^18^F]FPEB synthesized by radiochemists in the adjacent radiochemistry laboratory following the published methods [25] to be administered to participants in the scanning chambers.

#### 4.2.2. Statistical Analyses

Data for participants from IND were expressed as the standard uptake value ratio (SUVR) with the cerebellum as reference region because there is minimal radio tracer uptake in the cerebellum [26,54]. Assuming that there is no difference in nonspecific tracer binding between regions and between participant cohorts, we approximated nondisplaceable binding potentials (BP_ND_) [17,26] as the (SUVR-1) [36] for participants from IND (Table 2).

Data for participants from JHU were represented as regional nondisplaceable binding potentials (BP_ND_s) [14,17,23,26] by reference tissue graphical analysis (RTGA) [55] with the cerebellar white matter as the reference region [14,17,25,26,54].

Due to the small sample size we expressed the results for the combined cohorts from IND and JHU as dot plots with box plots representing descriptive statistics utilizing R (R Foundation, Vienna, Austria) [30].

After constructing the plots of our data, several group differences were observed across the regions tested. To confirm the effect of group status (e.g., FXS versus TD versus IASD), we used analysis of variance (ANOVA) utilizing R (R Foundation, Vienna, Austria) [30], using group and region as the main factors, with age and sex as covariates. As the ANOVA showed evidence of a significant effect of group on mGluR_5_ uptake, we then conducted post hoc pairwise comparisons with Tukey’s Honest Standard Differences (HSD) utilizing R (R Foundation, Vienna, Austria) [30,31]. HSD was chosen as the more traditional Bonferroni correction lacked statistical power given our smaller sample size. 

## 5. Conclusions

We confirmed our earlier finding of reduced cerebral mGluR_5_ expression [17] in a sample of men with FXS in contrast to participants with IASD and TD of both sexes. The significantly reduced mGluR_5_ expression in all brain regions of men with FXS provides a possible molecular basis for the neurobehavioral phenotype of individuals with FXS [10]. Reduced cortical mGluR_5_ expression may provide a basis for the cognitive deficits (delayed socialization) of individuals with FXS [56]. Reduced limbic mGluR_5_ expression may provide a basis for the avoidance behaviors of individuals with FXS [56].

We showed increased cortical cerebral mGluR_5_ expression in participants of both sexes with IASD in contrast to participants with TD and men with FXS. Since all participants with IASD were recruited initially for studies of children with high-functioning autism, the increased cortical cerebral mGluR_5_ expression may represent a molecular feature of IASD or of superior intelligence.

The proposed protocol may provide a biomarker for measurement of mGluR_5_ expression for clinical trials of FXS and other subtypes of ASD. The proposed protocol may provide a tool to utilize precision medicine for diagnostic and therapeutic interventions for ASD and related conditions.

## Figures and Tables

**Figure 1 ijms-22-02863-f001:**
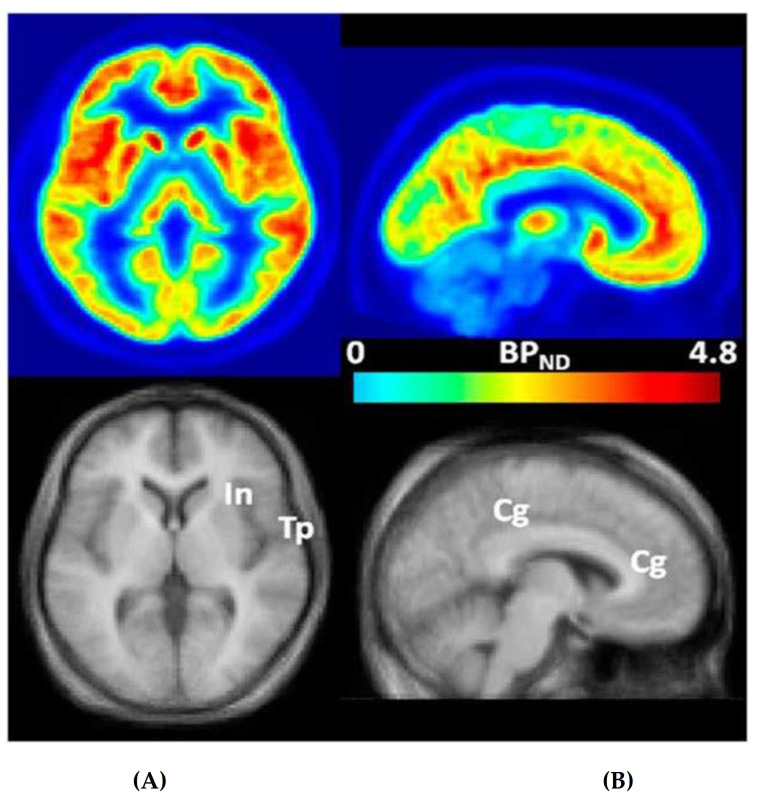
Transaxial (**A**) and sagittal (**B**) non-displaceable binding potential (BP_ND_) [27] images of 3-[^18^F]fluoro-5-(2-pyridinylethynyl)benzonitrile ([^18^F]FPEB) (top) and matching magnetic resonance (MR) images (bottom) in statistical parametric mapping (SPM) [25] standard space. Regions with high BP_ND_ values, name.ly insular (In), temporal (Tp), and cingulate (Cg) cortices, are indicated on co-registered MRimages [25]. This research was originally published in *JNM*. Wong DF, Waterhouse R, Kuwabara H, Kim J, Brašić JR, Chamroonrat W, Stabins M, Holt DP, Dannals RF, Hamill TG, Mozley PD. ^18^F-FPEB, a PET radiopharmaceutical for quantifying metabotropic glutamate 5 receptors: a first-in-human study of radiochemical safety, biokinetics, and radiation dosimetry. J Nucl Med. 2013;54:388-396. © SNMMI [25].

**Figure 2 ijms-22-02863-f002:**
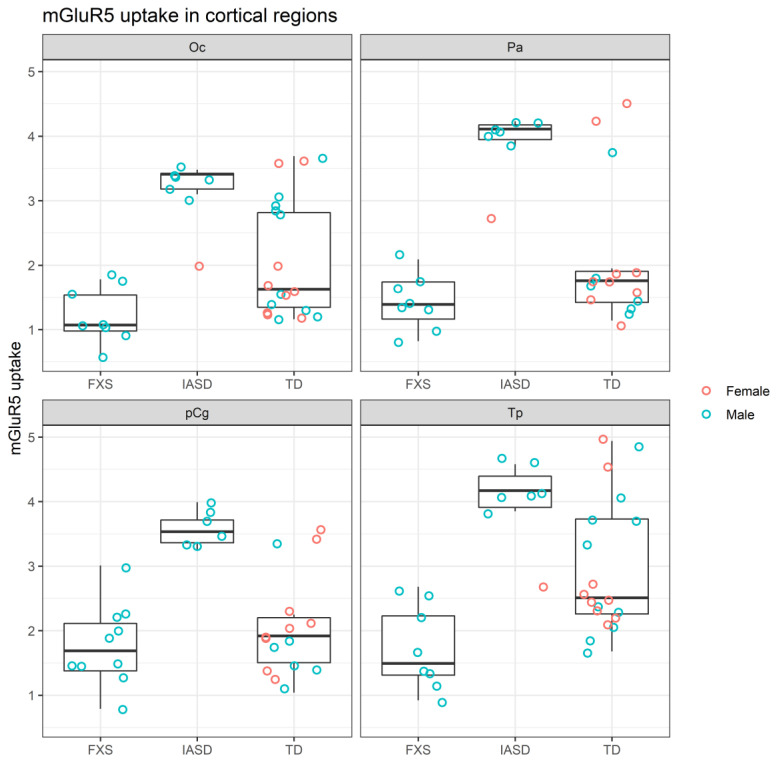
mGluR_5_ uptake in cortical regions of participants grouped by cohort [26,30]. FXS: Fragile X syndrome; IASD: Idiopathic autism spectrum disorder; mGluR_5_: Metabotropic glutamate receptor subtype 5; Oc: Occipital cortex; Pa: Parietal cortex; pCg: Posterior cingulate cortex; TD: Typical development; Tp: Temporal cortex.

**Figure 3 ijms-22-02863-f003:**
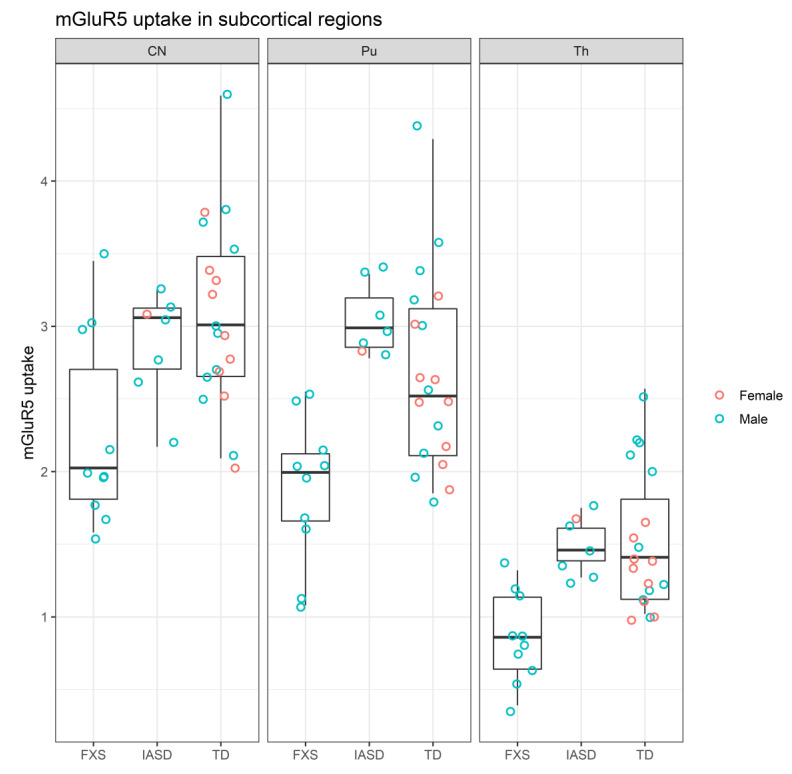
mGluR_5_ uptake in subcortical regions of participants grouped by cohort [26,30]. CN: Caudate nucleus; FXS: Fragile X syndrome; IASD: Idiopathic autism spectrum disorder; mGluR_5_: Metabotropic glutamate receptor subtype 5; Pu: Putamen; TD: Typical development; Th: Thalamus.

**Table 1 ijms-22-02863-t001:** Clinical characteristics of participants [26,29].

Participant	Group	Institution	Sex	Age in Years	BMI
INDTD01	TD	IND	Male	44	.
INDTD02	TD	IND	Male	57	.
INDTD07	TD	IND	Female	62	.
INDTD08	TD	IND	Female	62	.
INDTD14	TD	IND	Male	28	25.8
INDTD16	TD	IND	Male	31	29.8
INDTD17	TD	IND	Male	38	25.0
INDTD30	TD	IND	Female	28	.
INDTD35	TD	IND	Female	56	42.3
INDTD47	TD	IND	Female	22	.
INDTD48	TD	IND	Female	29	.
INDTD49	TD	IND	Female	20	.
JHUTD4	TD	JHU	Female	19	28.4
JHUTD6	TD	JHU	Female	19	.
JHUTD14	TD	JHU	Male	24	21.7
JHUTD105	TD	JHU	Male	26	.
JHUTD1001	TD	JHU	Male	32	27.1
JHUTD1002	TD	JHU	Male	27	28.6
JHUTD1005	TD	JHU	Male	39	29.3
JHUASD3	IASD	JHU	Male	18	28.8
JHUASD4	IASD	JHU	Male	18	.
JHUASD5	IASD	JHU	Male	19	22.2
JHUASD7	IASD	JHU	Female	18	22.3
JHUASD8	IASD	JHU	Male	23	28.5
JHUASD9	IASD	JHU	Male	20	19.4
JHUASD12	IASD	JHU	Male	22	20.7
INDFXS34	FXS	IND	Male	23	36.6
INDFXS38	FXS	IND	Male	24	30.9
INDFXS40	FXS	IND	Male	22	33.2
INDFXS41	FXS	IND	Male	27	25.8
INDFXS42	FXS	IND	Male	34	.
INDFXS44	FXS	IND	Male	26	24.1
INDFXS45	FXS	IND	Male	33	22.0
INDFXS-M50	FXS	IND	Male	57	34.1
JHUFXS2	FXS	JHU	Male	24	34.9
JHUFXS4	FXS	JHU	Male	27	28.3

BMI: Basal metabolic index; FXS: Fragile X syndrome; IASD: idiopathic autism spectrum disorder; IND: Institute for Neurodegenerative Disorders; JHU: Johns Hopkins University; TD: Typical development; (Period): Missing data.

**Table 2 ijms-22-02863-t002:** Metabotropic glutamate receptor subtype 5 uptake in regions of interest of participants [26,29].

Participant	CN	mTp	Oc	Pa	pCg	Pu	Th	Tp
INDTD01	3.01	1.94	1.41	1.72	1.92	2.52	1.08	2.37
INDTD02	3.03	2.05	1.63	1.76	1.04	2.35	1.30	2.35
INDTD07	3.31	2.04	1.50	1.76	2.07	2.70	1.60	2.56
INDTD08	3.01	2.02	1.57	1.83	1.97	2.64	1.32	2.50
INDTD14	2.12	1.41	1.16	1.43	1.72	1.85	1.06	1.68
INDTD16	2.68	1.61	1.20	1.26	1.42	2.03	2.24	1.84
INDTD17	2.63	1.82	1.37	1.36	1.55	2.03	1.16	2.04
INDTD30	2.42	1.73	1.26	1.14	1.26	1.87	1.08	2.17
INDTD35	3.82	1.90	1.74	1.86	2.15	2.48	1.43	2.39
INDTD47	2.09	1.29	2.17	1.41	1.86	1.95	1.02	2.74
INDTD48	3.30	1.76	1.94	1.95	2.25	2.49	1.51	2.51
INDTD49	2.75	1.57	1.32	1.56	1.46	2.19	1.07	2.10
JHUTD4	2.80	3.19	3.48	4.31	3.39	3.08	1.43	4.54
JHUTD6	3.38	3.57	3.69	4.48	3.65	3.16	1.32	4.94
JHUTD14	2.47	3.24	3.15	3.73	3.44	2.94	1.41	4.09
JHUTD105	3.73	.	2.86	.	.	3.30	2.11	3.73
JHUTD1001	4.59	.	3.64	.	.	4.29	2.57	4.86
JHUTD1002	3.83	.	2.77	.	.	3.66	2.21	3.73
JHUTD1005	3.58	.	2.77	.	.	3.24	2.02	3.26
JHUASD3	2.17	3.06	3.26	4.05	3.43	3.87	1.27	3.97
JHUASD4	2.62	3.43	3.10	4.17	3.99	3.03	1.45	3.85
JHUASD5	2.79	3.25	3.48	3.84	3.25	2.78	1.46	4.17
JHUASD7	3.13	.	2.03	2.74	.	2.84	1.67	2.72
JHUASD8	3.12	3.54	3.42	4.18	3.74	3.36	1.55	4.21
JHUASD9	3.06	3.35	3.43	4.23	3.64	3.36	1.32	4.58
JHUASD12	3.25	2.46	3.41	4.11	3.34	2.99	1.75	4.58
INDFXS34	1.96	1.00	1.06	1.42	1.37	2.01	0.83	1.40
INDFXS38	1.58	0.69	0.59	0.82	0.79	1.08	0.61	0.92
INDFXS40	1.65	0.81	0.82	0.96	1.40	1.13	0.39	1.14
INDFXS41	2.14	1.25	1.08	1.36	1.56	1.65	0.56	1.59
INDFXS42	3.45	2.24	1.78	2.09	2.18	2.55	1.25	2.68
INDFXS44	1.76	1.01	1.03	1.23	1.23	1.69	0.73	1.37
INDFXS45	2.89	1.74	1.47	1.73	1.91	2.16	1.32	2.13
INDFXS-M50	2.99	2.24	1.73	1.77	1.82	2.50	1.19	2.53
JHUFXS2	2.05	2.71	.	.	3.01	2.01	0.97	.
JHUFXS4	2.00	2.21	.	2.7	2.25	1.98	0.89	.

CN: Caudate nucleus; mTp: Medial temporal cortex; Oc: Occipital cortex; Pa: Parietal cortex; pCg: Posterior cingulate cortex; Pu: Putamen; Th: Thalamus; Tp: Temporal cortex.

**Table 3 ijms-22-02863-t003:** Analysis of variance of mGluR_5_ uptake by group (FXS, IASD, and TD) and region [30].

Analysis of Variance by Group Status and Region			
Region	Degrees of freedom	Test statistic	Probability
Caudate nucleus	2	6.77	0.00364
Occipital cortex	2	12.8	0.00010
Parietal cortex	2	16.2	0.00003
Posterior cingulate cortex	2	14.6	0.00006
Putamen	2	10.1	0.00043
Thalamus	2	10.3	0.00038
Temporal cortex	2	12.3	0.00014

FXS: Fragile X syndrome; IASD: Idiopathic autism spectrum disorder; mGluR_5_: Metabotropic glutamate receptor subtype 5; TD: Typical development.

**Table 4 ijms-22-02863-t004:** *Post hoc* pairwise comparisons of mGluR_5_ uptake by group (FXS, IASD, and TD) and region [30,31].

Post hoc Pairwise Comparisons by Tukey’s Honest Standard Differences [30,31]				
Region	Pairwise Comparison	Adjusted Mean Difference	Standard Error	Probability
Caudate nucleus	FXS-IASD	−0.81837	0.301489	0.028089
	TD-IASD	−0.02072	0.298712	0.997341
	TD-FXS	0.797653	0.259542	0.011671
Occipital cortex	FXS-IASD	−1.88121	0.409271	0.000218
	TD-IASD	−0.74904	0.384949	0.143727
	TD-FXS	1.132169	0.353062	0.008855
Parietal cortex	FXS-IASD	−2.31154	0.476181	0.000173
	TD-IASD	−1.56255	0.493592	0.010802
	TD-FXS	0.748987	0.462568	0.25626
Posterior cingulate cortex	FXS-IASD	−1.6965	0.362152	0.000203
	TD-IASD	−1.5125	0.425317	0.00414
	TD-FXS	0.184006	0.356537	0.863175
Putamen	FXS-IASD	−1.23401	0.29236	0.000529
	TD-IASD	−0.31288	0.289668	0.532202
	TD-FXS	0.921134	0.251683	0.002488
Thalamus	FXS-IASD	−0.69753	0.197012	0.003572
	TD-IASD	0.084401	0.195197	0.902066
	TD-FXS	0.781931	0.169601	0.000214
Temporal cortex	FXS-IASD	−2.18986	0.490827	0.000294
	TD-IASD	−0.75798	0.461658	0.243837
	TD-FXS	1.431881	0.423417	0.005678

FXS: Fragile X syndrome; IASD: Idiopathic autism spectrum disorder; mGluR^5^: Metabotropic glutamate receptor subtype 5; TD: Typical development.

## Data Availability

The data presented in this study are openly available in [Zenodo]. Available online: https://doi.org/10.5281/zenodo.4395102 (accessed on 6 March 2021) [26].

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
