# Peer review of "Cerebral Expression of Metabotropic Glutamate Receptor Subtype 5 in Idiopathic Autism Spectrum Disorder and Fragile X Syndrome: A Pilot Study"

_ijms, 2021, doi:10.3390/ijms22062863_

Round 1

Reviewer 1 Report

The paper written by the following Authors: James R. Brašić, Ayon Nandi, David S. Russell, Danna Jennings, Olivier Barret, Samuel D. Martin, Keith Slifer, Thomas Sedlak, John P. Seibyl, Dean F. Wong and Dejan B. Budimirovic , entitled “Elevated Cerebral Expression of Metabotropic Glutamate Receptor Subtype 5 in Autism Spectrum Disorder ” presents an interesting study on a autism spectrum disorder .

Although the paper is interesting, I have some major concerns:

Title

The title reflects the results presented here.

Abstract

The abstract is lacking the aim of the material and methods description as well as an informative conclusion. It should be written in more details.

Introduction

There is no aim of the study. It should be included in the manuscript.

Material and Methods

Thee is no information about the statistical methods applied in this manuscript.

Results

Where physical conditions included in the analyzed time of operation (Table 4)?

Discussion

In the discussion part there is no limitation to the studies. It should be included in the manuscript.

Conclusions

There is no informative conclusions. It should be extended with data from the results part.

Reviewer 2 Report

Dear Authors,

your article entitled "Elevated cerebral expression of metabotropic glutamat receptor subtype 5 in autism spectrum disorder" provides an interesting study focused on the expression of mGluR5 in vivo in a cohort of 19 TDI (10 males and 9 females), 7 ASD (6 males and 1 female) and 10 FXS males individuals. Despite a great interest on this topic  due to the need of development of specific outcome measures particularly during clinical trials, I suggest some major revisions:

-there is an excessive and inappropriate self-citations (Some ref.s such as Bear et al., 2004 and others are  not mentioned; furthermore in this paper 56 citations are too many);

-I would avoid to write "larger sample" referring to patients included in the study. No statistical analysis has been performed and the cohort of patients/individuals included in the study is too small (as mentioned by the authors in the "Limitations");

-I encourage to compact figures in two, i.e. Figures 2, 3, 4, S3 and 5 should be outlined in a single figure and Figures 6, 7 and 8 in a second one;

-Figure S1 and S2 are redundant, these results could be expressed in number in table S1, that may be included in the main text.

Minor revisions.

-page 2 "Introduction" last line: change "wit FXS" in "with FXS";

-page 12 "Discussion" line 14: add a space between "regionf of participants".

Reviewer 3 Report

This is a clinically important topic considering the lack of validated outcome measures and biomarkers for ASD and FXS. It would contribute to the field if the authors could discuss possible reasons for the opposite results in ASD versus FXS and the viability of mGluR5 inhibitors as a therapeutic strategy for these disorders.

The first sentence of the discussion: I believe the authors meant to say, “compared to participants of both…” rather than “and participants of both….”.

Round 2

Reviewer 1 Report

I accept.

Reviewer 2 Report

Dear Authors,

in the present form your article is suitable for publication in International Journal of Molecular Sciences.